# Chloroplast Genomes and Phylogenetic Analysis of Three *Carthamus* (Asteraceae) Species

**DOI:** 10.3390/ijms242115634

**Published:** 2023-10-26

**Authors:** Tiange Yang, Saimire Aishan, Jiale Zhu, Yonghua Qin, Jiao Liu, Hong Liu, Jun Tie, Jiangqing Wang, Rui Qin

**Affiliations:** 1Hubei Provincial Key Laboratory for Protection and Application of Special Plant Germplasm in Wuling Area of China, College of Life Sciences, South-Central Minzu University, Wuhan 430074, China; yangtge@163.com (T.Y.);; 2College of Computer Science, South-Central Minzu University, Wuhan 430074, China

**Keywords:** *Carthamus*, chloroplast genome, angiosperms353, phylogeny

## Abstract

The genus *Carthamus* Linnaeus, which belongs to the tribe Cardueae in the Asteraceae family, originated in the Mediterranean region and consists of approximately 20 species worldwide. Understanding the phylogeny of the *Carthamus* is crucial for the cultivation of *C. tinctorius*. Although chloroplast genomes are widely used for species identification and evolutionary studies, there have been limited investigations on the chloroplast genomes of *Carthamus* species. In this study, we assembled the chloroplast genomes of *C. persicus*, *C. tinctorius* × *C. persicus,* and *C. lanatus* and combined them with the five chloroplast genomes of *C. tinctorius* for comparative genomic analysis. The sizes of the chloroplast genomes of *C. lanatus*, *C. persicus*, and *C. tinctorius* × *C. persicus* were 152,602 bp, 153,177 bp, and 153,177 bp, respectively. Comparative analysis showed that the chloroplast genome structures of the four *Carthamus* species were highly conserved. Additionally, the phylogenomic analysis demonstrated that the plastid genome and angiosperms353 dataset significantly improved the phylogenetic support of *Carthamus* species. This analysis supported *Carthamus* as a monophyletic taxon and its internal division into the sect. *Carthamus* and sect. *Atractylis*. The *Carthamus* was closely related to *Carduncellus*, *Femeniasia*, *Phonus*, and *Centaurea*. In conclusion, this study not only expands our understanding of the cp genomes of *Carthamus* species but also provides support for more comprehensive phylogenetic studies of *Carthamus*.

## 1. Introduction

Chloroplasts, which are organelles exclusive to plants, play an essential role in indirectly providing significant amounts of nutrients and oxygen to humans through photosynthesis [1,2]. Moreover, they have the ability to effectively bind carbon dioxide, thus aiding in mitigating global warming. The chloroplast (cp) genome has served as a valuable source of molecular data for investigating plant phylogeny and evolution; this is attributed to its predominantly maternal inheritance, although patrilineal [3,4,5] or biparental inheritance [6,7,8,9] can occur in certain species and its relatively conserved nature, which differs from plant mitochondrial genomes [10]. In most plants, the cp genomes possess a cyclic quadripartite structure with two single-copy regions separated by two inverted repeat regions, although linear structures are found in some species [11,12,13]. Recent advancements in sequencing technologies have facilitated whole-genome sequencing and large-scale phylogenetic analyses of numerous plants, resulting in an enhanced understanding of phylogeny [14,15]. Consequently, an increasing number of plant cp genomes have been reported, significantly contributing to our knowledge of plant phylogeny and aiding in plant taxonomy.

Targeted enrichment of nuclear genes addresses the potential limitation of relying solely on organelle genome data for estimating phylogeny. Unlike large nuclear genomes, angiosperm organelle genomes are predominantly maternally inherited and are prone to high rates of genetic drift in small populations, resulting in potentially less accurate estimates of lineage divergence [16]. The Angiosperms353 dataset has gained widespread usage in angiosperm phylogeny [17]. This dataset comprises 353 targeted low-copy nuclear genes that are highly representative and have become a valuable tool for molecular systematics and population genetics [17,18,19]. Additionally, researchers can easily access the dataset through enrichment sequencing (Hyb-Seq), transcriptome sequencing (RNA-Seq), and genome skimming to obtain the corresponding data [20]. Previous studies have demonstrated the feasibility of applying the angiosperms353 dataset in phylogenetic studies of Asteraceae [21].

The *Carthamus* Linnaeus, belonging to the tribe Cardueae of the family Asteraceae, originated in the Mediterranean region and comprises approximately 20 species worldwide [9,10,11]. These *Carthamus* species are predominantly found in arid and semi-arid environments and exhibit high tolerance to drought stress [22]. Among them, *C. tinctorius* Linnaeus is the sole domesticated oilseed crop of economic importance, while the remaining species, including *C. lanatus* Linnaeus and *C. persicus* Desf. ex Willd., are wild weeds distributed from northwestern India to the Mediterranean Sea and its surrounding regions [23]. Previous phylogenetic studies on the *Carthamus* primarily utilized plastid and nrITS (nuclear ribosomal internal transcribed spacer) markers. Bowles et al. conducted an investigation of the phylogenetic relationships within *Carthamus* using nrITS, plastid, and microsatellite markers [24]. Their results revealed that the *Carthamus* consisted of two sections: sect. *Carthamus* and sect. *Atractylis*. To objectively evaluate the *Carduncellus*-*Carthamus* complex, Vilatersana et al. performed a phylogenetic analysis using nrITS and plastid markers, highlighting the subdivision of the complex into four genera: *Carduncellus*, *Carthamus*, *Phonus*, and *Femeniasia* [25]. Despite the widespread use of cp genomes in species identification and evolutionary studies, limited studies have been carried out on the cp genome of *Carthamus* species. Currently, only the cp genome of *C. tinctorius* has been reported [26,27], which hinders phylogenetic studies of *Carthamus*.

Here, we assembled the cp genomes of *C. persicus*, *C. tinctorius* × *C. persicus,* and *C. lanatus* and performed a comparative genomic analysis with the known cp genome of *C. tinctorius* to explore differences in these cp genomes. In addition, we investigated the phylogenetic relationships between these four *Carthamus* species and other Cardueae species. This study not only lays the foundation for understanding the cp genomes of *Carthamus* but also provides data support for more comprehensive phylogenetic studies of *Carthamus*.

## 2. Results

### 2.1. Basic Characteristics of Chloroplast Genome

The sizes of the cp genome of *C. lanatus*, *C. persicus*, *C. tinctorius* × *C. persicus*, and *C. tinctorius* were determined to be 152,602 bp, 153,177 bp, 153,177 bp, and 152,938–153,114 bp, respectively (Figure 1a, Appendix A). It is worth noting that *C. tinctorius* × *C. persicus*, which was a hybrid of *C. tinctorius* (male parent) and *C. persicus* (female parent), had the same cp genome size as its female parent at 153,177 bp (Figure 1b). Their cp genome structures displayed a typical quadripartite structure, consisting of a large single copy region (LSC), a small single copy region (SSC), and two inverted repeats (IR) regions. The differences in GC content among the four species were relatively small. However, it was observed that the IR region had a higher GC content compared to the LSC and SSC regions. In terms of gene composition, the number and type of genes were consistent among the four *Carthamus* species. A total of 131 genes were identified, including 86 protein-coding genes (PCGs), 8 rRNAs, and 37 tRNAs. Among these genes, eighteen genes had two copies. These cpPCGs were mainly classified into five categories (Appendix A): five photosystem I genes, fifteen photosystem II genes, six subunits of ATP synthases, nine large ribosomal proteins, and twelve small ribosomal proteins. In addition, 14 genes (*ndh*A, *ndh*B, *atp*F, *pet*B, *pet*D, *rpo*C1, *rpl*2, *rpl*16, *trn*I-GAU, *trn*G-GCC, *trn*L-UAA, *trn*V-UAC, *trn*A-UGC, *trn*K-UUU) contained one intron, and three genes (*clp*P, *rps*12, *ycf*3) contained two introns.

### 2.2. Repeats Identification

Based on the comparison of intraspecific and interspecific repetitive sequences in the cp genomes of four *Carthamus* species, it was found that they had similar types of repeats but with some differences in terms of their numbers (Figure 2a, Appendix A). In the four species, namely *C. tinctorius*, *C. persicus*, *C. lanatus*, and the hybrid species *C. tinctorius* × *C. persicus*, the differences in the number of simple sequence repeats (SSRs) were minimal. For instance, only mono-, di-, and trinucleotides were detected in *C. tinctorius*, *C. persicus*, and *C. tinctorius* × *C. persicus*, while *C. lanatus* had the additional presence of one tetranucleotide. Both *C. persicus* and *C. tinctorius* × *C. persicus* contained 36 SSRs, including 20 mononucleotides, 11 dinucleotides, and 5 trinucleotides. The *C. lanatus* contained 40 SSRs, including 21 mononucleotides, 11 dinucleotides, 7 trinucleotides, and 1 tetranucleotide. Further analysis was conducted specifically on the five *C. tinctorius* species. It was found that the differences in mononucleotides were more pronounced compared to other repeat types. Additionally, the distribution of SSRs revealed that they were primarily located in the LSC region, with a smaller number found in the SSC region. These SSRs were mainly distributed in the intergenic spacer regions. Moreover, the analysis of long repeats indicated that they predominantly consisted of forward, reverse, and palindromic repeats. These long repeats were mainly distributed in the LSC and Inverted Repeat (IR) regions. Among these long repeats, the largest proportion was accounted for by forward and palindromic repeats, while reverse repeats were the least common (Figure 2a, Appendix A).

### 2.3. Sequence Variation and IR Boundary Analysis

The genomic collinearity analysis revealed that the four cp genomes exhibited a relatively conserved pattern with no significant rearrangements (Appendix A); this suggests that the organization of genes and other functional elements in the cp genomes of four *Carthamus* species was similar. To further understand the differences in the cp genomes of four *Carthamus* species, we performed a sequence difference visualization analysis (Figure 3a, Appendix A). In interspecific comparisons, we found that *C. lanatus* had more sequence variation compared to *C. tinctorius* and *C. persicus*. On the other hand, *C. tinctorius* and *C. persicus* had relatively few sequence differences, indicating a closer relationship between these two species. In intraspecific comparisons of *C. tinctorius*, different *C. tinctorius* species had some sequence variation, all of which occurred in intergenic spacer regions. IR boundary analysis revealed that the chloroplast boundaries of these *Carthamus* species were highly conserved, and the differences between gene locations and boundaries were smaller. For example, *trn*N and *rpl*2 genes were located within the IR regions, *ndh*F genes were located within the SSC regions, and the LSC-IRb and SSC-IRa boundaries were located on the *ycf*1 and *rps*19 genes, respectively.

### 2.4. Codon Usage Analysis

We selected 53 PCGs with a length greater than 300 base pairs for codon analysis. The analysis revealed variations in various aspects of codon composition, such as the number of codons, as well as the GC1, GC2, GC3, and overall GC content, among the 19 species of the tribe Cardueae (Appendix A). Our results indicated that the content of GC1 and GC2 was higher compared to that of GC3 and the overall GC content; this suggested that the distribution of GCs among codons was not uniform. The trend in GC1, GC2, and GC3 remained consistent across the 19 Cardueae species. Specifically, leucine was found to be the most abundant amino acid in these cp genomes, while cysteine was the less frequent amino acid. Within the 59 codons, the more common isoleucine was encoded by AUU, and the less common cysteine by UGC (Appendix A). Additionally, leucine encoded by CUC had the lowest RSCU value, whereas leucine encoded by UUA had the highest RSCU value. Both Tryptophan (UGG) and methionine (AUG) were encoded by a single codon and had RSCU values of 1. Furthermore, 29 codons had RSCU values greater than 1, indicating biased use (Figure 4).

### 2.5. Phylogenetic Analysis

To determine the phylogenetic position of the four *Carthamus* species, we constructed phylogenetic trees by combining cp genome, nrITS, and angiosperm 353 data for Cardueae species. Of these, four species (*Phonus arborescens*, *Femeniasia balearica*, *Carduncellus pinnatus*, *Carduncellus rhaponticoides*) acquired only about 70 PCGs. Additionally, we also assembled the angiosperms353 data from eight species (*P. arborescens*, *F. balearica*, *C. pinnatus*, *C. rhaponticoides*, *C. persicus*, *C. tinctorius*, *C. lanatus*, *C. tinctorius* × *C. persicus*). The average angiosperms353 gene number was 333, with a minimum gene number of 303. We performed informative loci and tree-building modeling statistics for the three sequence matrices (Appendix A). The AP353 matrix had a length of 425,889 bp, with 103,034 parsimony informative sites, 191,242 variable sites, and 147,787 conserved sites. The cpPCGs matrix length was 79,671 bp, consisting of 13,032 parsimony informative sites, 33,016 variable sites, and 39,417 conserved sites. The nrITS matrix length was 830 bp, with 196 parsimony informative sites, 310 variable sites, and 402 conserved sites. The phylogenetic trees of cpPCGs and AP353 matrixes exhibited high support, while the nrITS phylogenetic tree showed some branches with low support (Figure 5). Nevertheless, the overall structure remained similar across most branches of the three phylogenetic trees. These trees indicated that the *Carthamus*-*Carduncellus* complex was monophyletically sister to Centaurea in the Centaureinae branch. The *Carthamus* was closely related to *Phonus*, *Femeniasia*, and *Carduncellus*. Our results were consistent with previous studies, which divided *Carthamus* into two sections, *C. persicus,* and *C. tinctorius*, along with their hybrids in sect. *Carthamus*, and *C. lanatus* in sect. *Atractylis*. Based on three phylogenetic trees, we observed conflicts between the nucleoplasm. In the plastid phylogeny, *C. rhaponticoides* did not cluster with *C. pinnatus* but showed a closer relationship to *F. balearica*. In the AP353 and nrITS phylogenetic trees, *C. rhaponticoides* clustered with *C. pinnatus*. Additionally, nucleoplasmic differences were also identified in other genera.

## 3. Discussion

### 3.1. Chloroplast Genome Structure Variation within Three Carthamus Species

In this study, we assembled and analyzed the cp genomes of *C. persicus*, *C. tinctorius* × *C. persicus*, and *C. lanatus* for the first time. We found that these cp genomes shared a similar size and structure with the previously reported *C. tinctorius*, all exhibiting a typical quadripartite structure. While there have been instances of evolutionary events such as genome rearrangements, gene deletions, IR contractions, and expansions, plastids generally demonstrate a high level of conservation in terms of genome size, structure, and gene content [28]. Our study revealed that the cp genomes of four *Carthamus* species showed minimal variability in size and were conserved in terms of genome structure, gene composition, and gene order. In terms of sequence differences, *C. lanatus* exhibited significant variation compared to the other three *Carthamus* species, displaying more sequence variation. In addition, due to maternal inheritance, the cp genome of the female parent (*C. persicus*) was inherited by *C. tinctorius* × *C. persicus*, resulting in identical cp genomes. The selfed progeny of this hybrid also possessed the same chloroplast genomes. The GC content in the IR region was significantly higher than that in the LSC and SSC regions, which was a common characteristic in the cp genomes of most species [29,30]. Furthermore, the four *Carthamus* species showed consistency in the number of total genes, PCGs, rRNA, and tRNA genes, suggesting that their cp genomes were highly similar, which partly explains the similarity in their morphological features. The best synonymous codons in the four *Carthamus* species mostly end in A or U, resulting in increased AT content in the genomes; this supports the A/T codon bias prevalent in the cp genomes of higher plants [29]. Plant cp genomes have a high number of SSRs, which provide rich genetic information and can be used as important molecular markers to study species’ genetic evolution [31]. Among the four Carthamus species, pentanucleotides and hexanucleotides were not detected, while tetranucleotides were only found in *C. lanatus*. Dynamic changes in the IR region may lead to cp genomic variation in angiosperms. The IR boundaries were highly conserved in the four *Carthamus* species, suggesting that they are closely related. However, the complete cp genomes of *Carduncellus*, *Phonus*, and *Femeniasia* were still missing in this study for comparison with *Carthamus*. In the future, more material from the *Carthamus* and its relatives should be collected to obtain genomic data and further investigate the differences between them.

### 3.2. Phylogeny of Carthamus Species

Phylogenetic trees for Cardueae species were constructed using Maximum Likelihood and Bayesian Inference methods. The topology of the phylogenetic trees obtained by these two methods was similar. The phylogenetic trees of the cpPCGs and AP353 matrices showed high support. However, the use of nrITS sequences alone did not provide strong support for some species, possibly due to the limited number of informative sites in nrITS sequences. Previous phylogenetic studies have suggested that the *Carthamus*-*Carduncellus* complex was divided into four genera (i.e., *Carthamus*, *Carduncellus*, *Phonus*, and *Femeniasia*) and that the branch of the genus *Carthamus* could be divided into sect. *Carthamus* and sect. *Atractylis*. Our phylogenetic trees indicated that the complex was closely related to *Centaurea*. Within the complex, the four *Carthamus* species were clustered into a single clade, suggesting that *Carthamus* was monophyletic. Additionally, the branching structures between the cpPCGs and the AP353 phylogenetic trees were not completely consistent. The plastid phylogeny showed that *C. rhaponticoides* was more closely related to *F. balearica*, which was not consistent with the nuclear phylogenetic trees. The nuclear phylogenetic trees showed *C. rhaponticoides* clustered with *C. pinnatus* and *F. balearica* clustered with *P. arborescens*, which was consistent with previous studies [24,25,31,32]. Within the *Carthamus*, our results also show that *C. persicus* clustered with *C. tinctorius* × *C. persicus* and *C. tinctorius*, with the three having a closer relationship, all belonging to sect. *Carthamus*, while *C. lanatus* belonged to sect. *Atractylis*.

Differences between the plastid phylogenetic tree and the nuclear phylogenetic tree may arise from various factors, including hybridization, incomplete lineage sorting (ILS), chloroplast capture, and genetic differences in plastids [33,34,35,36]. ILS is a common occurrence in phylogenies, where nucleoplasmic conflicts in Asteridae could be attributed to ancient chloroplast capture or ILS. Different DNA fragments exhibit varying evolutionary rates, leading to disparities between gene trees and species trees—the phenomenon of nucleoplasmic conflicts in Asteraceae warrants further investigation in the future. Hybridization is a frequent natural occurrence and is relatively common in Asteraceae, as exemplified by the hybridization and polyploid evolutionary radiation observed in *Achillea* [37]. Additionally, *Phalacrocarpum* has demonstrated susceptibility to frequent hybridization [38]; this is one of the reasons for the rich diversity of Asteraceae species. Nucleoplasmic conflicts can be affected by interspecific hybridization in angiosperms; this is due to the prevalence of maternal inheritance in angiosperms, resulting in the offspring inheriting the cp DNA of the female parent. Over time, the offspring of hybridization may evolve into a new species while still retaining the cp DNA of the female parent. Although most eukaryotic organelle genomes are maternally inherited, there are cases of biparental inheritance of plastids. Previous studies have indicated that around 20% of angiosperm pollen germ cells contain chloroplasts, suggesting the possibility of biparental inheritance of plastids [6,7]. For example, the *Oenothera* [9] and *Pelargonium* [8] exhibit biparental inheritance of cp genomes. Additionally, paternal inheritance of plastids is more common in gymnosperms [3] and relatively less common in angiosperms, such as *Nicotiana tabacum* and *Arabidopsis thaliana* [4,5]. Previous studies have suggested that mild low-temperature stress promotes paternal inheritance of plastids [39], but this has not been reported in Asteraceae. *Carthamus* and its relatives are known for their preference for light and tolerance to drought, suggesting that low temperatures may not be the main factor contributing to paternal plastid inheritance. However, the specific influences and causes of paternal plastid inheritance are still unknown in the *Carthamus* and its relatives. Therefore, it is necessary to collect more *Carthamus* species and study the plastid inheritance pattern, hybridization, and ILS in order to explore a more accurate understanding of the phylogenetic network in the future.

## 4. Materials and Methods

### 4.1. Sampling, DNA Extraction, and Sequencing

The *C. lanatus* (specimen number: MH001) was cultivated in the experimental field at South-Central Minzu University, Wuhan, China. Fresh leaves of this species were collected for genome sequencing. Genomic DNA was extracted using the modified CTAB method [40]. A short insertion library was constructed, and then 2 × 150 bp paired-end reads were obtained from the BGI-Seq platform. Adaptors and low-quality reads were removed using Trimmomatic v. 0.39 [41]. The filtered reads were quality-controlled using Fastqc v. 0.11.9 [42]. In addition, short-read sequencing data of *C. tinctorius* (male parent, SRR3418170-SRR3418175), *C. persicus* (female parent, SRR3418173-SRR3418276), and *C. tinctorius* × *C. persicus* (hybrid offspring, SRR3418176-SRR3418167, SRR3418176-SRR3418269) [43] for cp genome assembly. Then, we also collected target-enriched genome sequencing data for *C. pinnatus* (SRR12917348), *C. rhaponticoides* (ERR5033745), *F. balearica* (ERR9230214), *P. arborescens* (ERR9230215), *Synurus deltoides* (SRR12849144), *Dolomiaea baltalensis* (SRR11926473), and *Xanthopappus subacaulis* (SRR12917358) from the NCBI database.

### 4.2. Chloroplast Genome Assembly and Annotation

To assemble the cp genomes of *C. tinctorius*, *C. lanatus*, *C. persicus*, and *C. tinctorius* × *C. persicus*, we used GetOrganelle v.1.7.5 [44] with the parameters: “-R 15 -t 20 -k 21, 45, 65, 85, 105 -F embplant_pt”. Subsequently, we assessed the integrity of the genomes using Bandage v. 0.8.1 software (http://rrwick.github.io/Bandage/ (accessed on 10 August 2023)). The cp genome annotation was performed using CPGAVAS2 [45] and PGA [46]. Additionally, we constructed the cp genome map using CPGview (http://www.1kmpg.cn/cpgview/ (accessed on 12 August 2023)) [47]. For the nrITS sequences, we also used GetOrganelle v.1.7.5 [44] with the parameter ‘-R 7 -t 20 -k 35, 85, 115 -F embplant_nr’ to assemble them. Then, these nrITS sequences were annotated using Geneious Prime v. 2022.2.2 [48], with *C. tinctorius* (KY397481) serving as the reference genome. Based on the target-enriched genome sequencing data of the *C. pinnatus*, *C. rhaponticoides*, *F. balearicus*, and *P. arborescens*, we used Geneious Prime v. 2022.2.2 [48] to assemble some chloroplast genes.

### 4.3. Repeat Structure Identification

Simple sequence repeats (SSRs) were identified using MISA v.2.1 software [49]. The minimum repeat numbers were set at 10, 6, 5, 5, 5, and 5 for mono-, di-, tri-, tetra-, penta-, and hexanucleotide, respectively. We used REPuter (https://bibiserv.cebitec.uni-bielefeld.de/reputer (accessed on 12 August 2023)) to predict the palindromic and complement, forward, and reverse repeats [50]. The minimum repeat size and Hamming distance were set to 30 and 3, respectively.

### 4.4. Codon Usage Analysis

To reduce sampling error, we excluded protein-coding genes (PCGs) shorter than 300 bp [51,52,53,54]. A total of 53 PCGs were used for codon usage analysis. We used CodonW v.1.4.4 [53,55] (https://codonw.sourceforge.net/culong.html#CodonW (accessed on 12 August 2023)) to calculate the GC content of the silent 1st, 2nd, and 3rd codon positions (GC1, GC2, and GC3), as well as the relative synonymous codon usage (RSCU) value. The RSCU value greater than 1 indicates a higher frequency of codon usage, while a value less than 1 indicates a lower frequency.

### 4.5. Sequence Variation Analysis

To analyze sequence variation in the cp genomes of the four *Carthamus* species, we performed multiple sequence alignments using MAFFT v.7.4 [56]. Then, aligned sequences were visualized using Geneious Prime v.2022.2.2 [48]. Additionally, We used mVISTA [57] to analyze sequence variation, with *C. tinctorius* (OR538395) serving as the reference genome.

### 4.6. Phylogenetic Analysis

We selected the cp genomes of the 19 species of the tribe Cardueae and *C. indicum* for the plastid phylogenetic analysis and used *Chrysanthemum indicum* as the outgroup (Appendix A). We extracted the 78 common cpPCGs from these 21 cp genomes and aligned them using MAFFT v.7.4 [56]. The aligned sequences were then concatenated using PhyloSuite v.1.2.3 [58] to create a cpPCGs matrix. Additionally, we compiled a nrITS matrix by combining the nrITS sequences of all 25 species. We also used easy353.py [20] to assemble the angiosperms353 dataset of the 11 species (*C. tinctorius*, *C. tinctorius* × *C. persicus*, *C. persicus*, *C. lanatus*, *C. pinnatus*, *C. rhaponticoides*, *F. balearica*, *P. arborescens*, *S. deltoides*, *D. baltalensis*, *X. subacaulis*). Then, we incorporated data from the public dataset (Kew Tree of Life Explorer, https://treeoflife.kew.org/ (accessed on 14 August 2023)) to create the angiosperms353 (AP353) matrix. Finally, we constructed phylogenetic trees for each of the three matrices (cpPCGs matrix, nrITS matrix, AP353 matrix) and performed comparative analysis.

The phylogenetic trees were constructed using Maximum Likelihood (ML) and Bayesian Inference (BI) methods. ModelFinder [59] was utilized to determine the best-fitting model for ML analysis. Use IQ-TREE v. 2.1.2 [60] to construct the ML tree with the parameter “-m MFP -bb 1000 -nt 30”. For the BI tree, MrBayes v. 3.2.6 [61] was used to obtain a maximum clade credibility (MCC) tree. The parameters for the BI analysis were set as follows: nst = 6, rates = invgamma. Bayesian inference was performed with the concatenated sequence, using one million generations, two runs, four chains, and 25% of trees were discarded as burn-in. Trees were sampled every 1000 generations. The resulting trees were visualized using Figtree v.1.4.3 (https://github.com/rambaut/figtree/releases (accessed on 16 August 2023)).

## 5. Conclusions

In this study, we conducted comparative genomic and phylogenetic analyses of the cp genomes of three *Carthamus* species. By conducting comparative genomic analyses, we were able to identify several findings. Firstly, our analysis revealed that the *Carthamus* species exhibited a high level of conservation in terms of genome structure, gene content, IR boundaries, repeats, and codon preference; this suggests that these species have maintained a relatively stable genomic framework throughout their evolution. However, we also observed some sequence variation in *C. lanatus*, suggesting that *C. lanatus* was more distantly related to *C. tinctorius* and *C. persicus*. To better understand the phylogenetic relationships of the three *Carthamus* species within the tribe Cardueae, we performed a phylogenomic analysis using the cp genomes, nrITS, and angiosperms353 dataset. This analysis greatly improved the support and resolution of the phylogenetic trees. Our phylogenomic analysis provided insights into the relationships between *Carthamus* and other closely related genera. We found that *Carthamus* was closely related to species such as *Carduncellus*, *Femeniasia*, *Phonus*, and *Centaurea*. This information contributes to our understanding of the relationships of these species. Overall, our study enriches the knowledge of the cp genomes of *Carthamus* species and provides a valuable reference for future research on their evolution and origin.

## Figures and Tables

**Figure 1 ijms-24-15634-f001:**
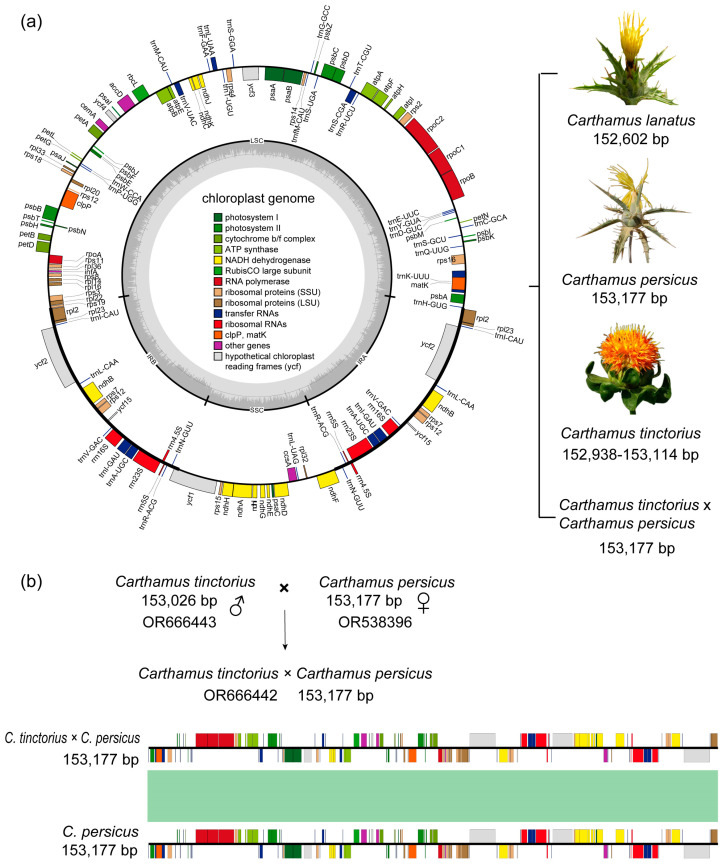
(**a**) The cp genome maps of *C. persicus*, *C. tinctorius*, *C. tinctorius* × *C. persicus* and *C. lanatus*. The genes are represented by colored boxes. Genes transcribed clockwise are shown inside the circle, while genes transcribed counter-clockwise are shown outside the circle. The small grey bars inside the circle indicate the GC content. (**b**) A schematic of the cross between *C. tinctorius* (male parent) and *C. persicus* (female parent) is shown, along with a comparison of the cp genome similarity between *C. persicus* and *C. tinctorius* × *C. persicus*. The green area in the center indicates a 100% colinear area. The colored boxes represent genes, as shown in Figure 1a.

**Figure 2 ijms-24-15634-f002:**
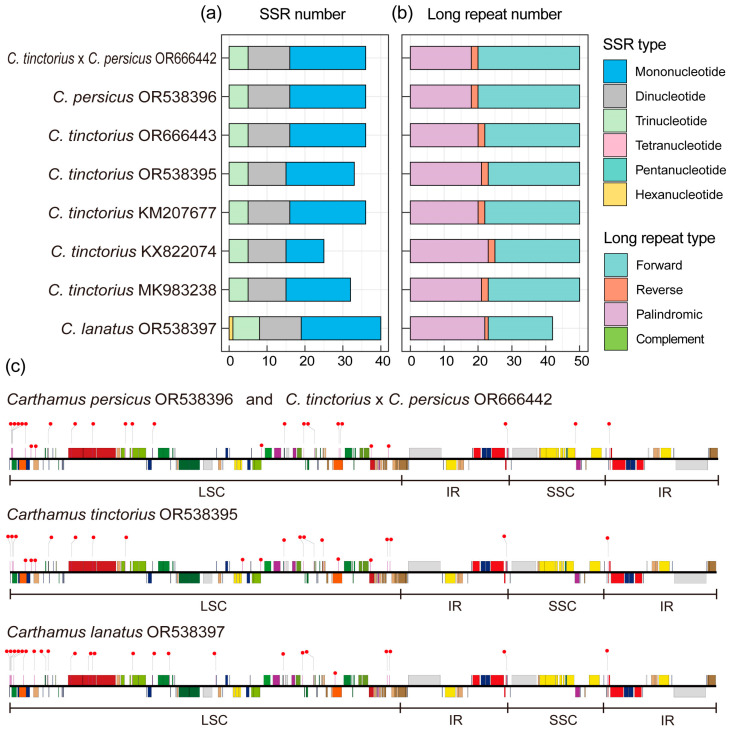
Repeat analysis on the cp genomes of four *Carthamus* species. (**a**) SSR statistics of three *Carthamus* species. Different types of SSRs are indicated by different colors. (**b**) Long repeat statistics of three *Carthamus* species. Different types of long repeats are indicated by different colors. (**c**) Distribution of SSRs on the cp genome of three *Carthamus* species. Red dots indicate SSR locations. The colored boxes represent genes, as shown in Figure 1a.

**Figure 3 ijms-24-15634-f003:**
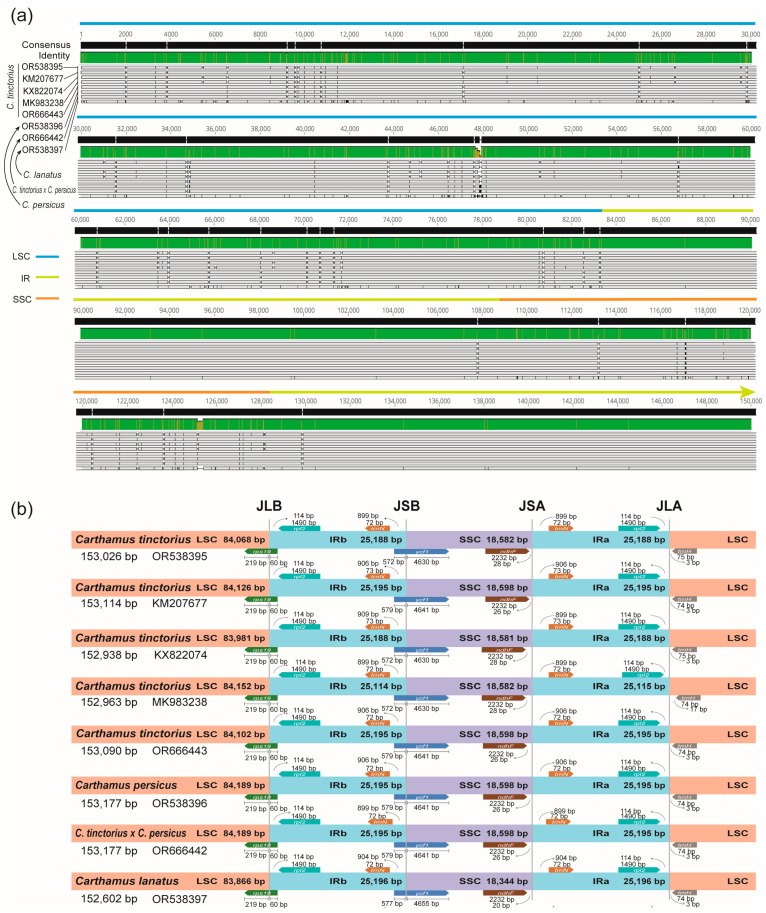
Variation analysis of cp genome. (**a**) Visual analysis of multiple sequence alignment in *C. tinctorius* and its relatives. (**b**) IR boundary analysis. Gene lengths in the corresponding regions were displayed above the boxes of gene names. The number of bp represented by the arrow showed genes away from a specific region of the cp genome. JLB (LSC/IRb), JSB (IRb/SSC), JSA (SSC/IRa), and JLA (IRa/LSC) denoted the junction sites between each corresponding two regions on the cp genome. The number of bp represented by the arrow showed genes away from a specific region of the cp genomes.

**Figure 4 ijms-24-15634-f004:**
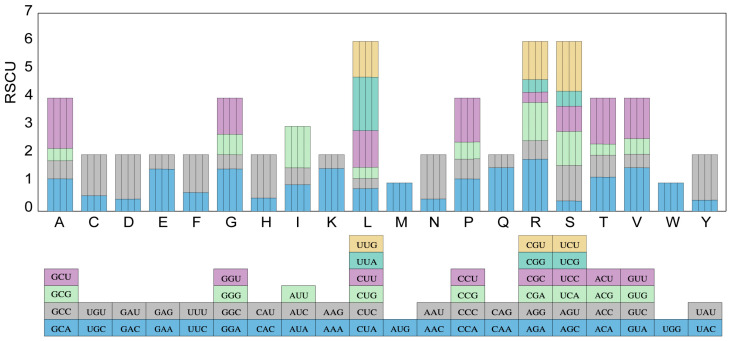
Relative synonymous codon usage (RSCU) analysis. Codon frequency of 20 amino acids in all PCGs of the cp genomes of four *Carthamus* species. The histogram above each amino acid shows codon usage. Colors in the column graph reflected codons in the same colors shown below the figure. A: alanine; R: arginine; N: asparagine; D: asparagine; C: cysteine; Q: glutamine; E: glutamic; G: glycine; H: histidine; L: leucine; I: isoleucine; K: lysine; M: methionine; F: phenylalanine; P: proline; S: serine; T: threonine; W: tryptophan; Y: tyrosine; V: valine. From left to right, *C. tinctorius* × *C. persicus*, *C. persicus*, *C. tinctorius*, and *C. lanatus*.

**Figure 5 ijms-24-15634-f005:**
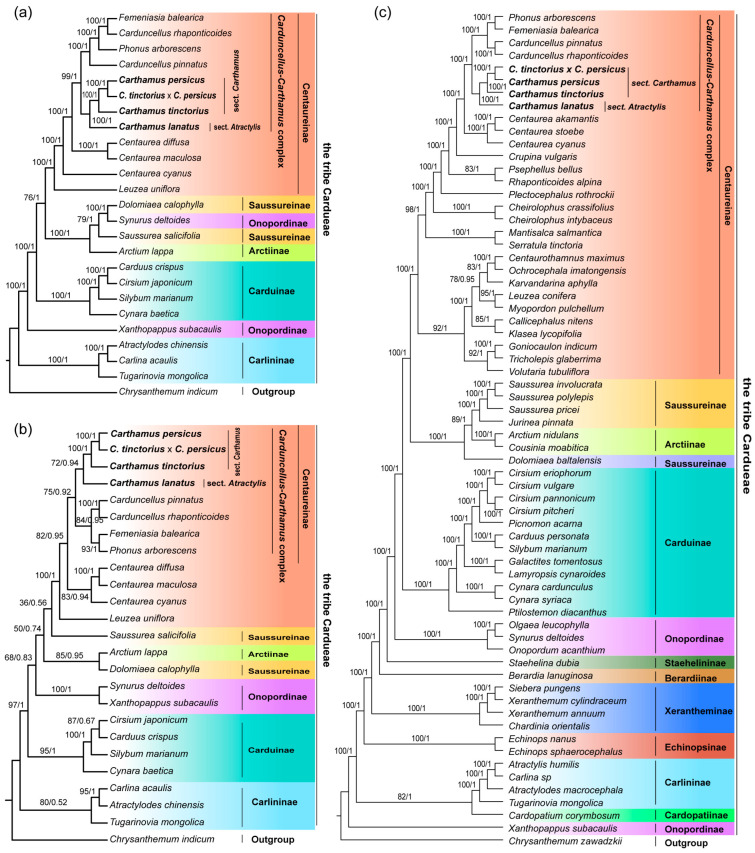
Phylogenetic trees of the tribe Cardueae using Maximum likelihood (ML) and Bayesian inference (BI) analyses. (**a**) Phylogenetic tree of cpPCGs matrix; (**b**) Phylogenetic tree of nrITS matrix; (**c**) Phylogenetic tree of AP353 matrix. The values on the nodes indicate the ML bootstrap values (left) and BI posterior probabilities (right). Different colors represent different subtribes.

## Data Availability

The data provided in this study are deposited in the NCBI GenBank database (https://www.ncbi.nlm.nih.gov/ (accessed on 25 August 2023)); the GenBank accession numbers are listed in Appendix A. Our data is also stored in the Youdao cloud note (https://note.youdao.com/s/8BsU3JBa (accessed on 30 August 2023)).

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
