# Peer review of "Chloroplast Genomes and Phylogenetic Analysis of Three Carthamus (Asteraceae) Species"

_ijms, 2023, doi:10.3390/ijms242115634_

Round 1
Reviewer 1 Report
Comments and Suggestions for Authors
1.Why not use the whole cp genomes for phylogenetic analysis?Suggest using whole genome data to analyze the evolutionary tree
2.'The phylogenetic trees were constructed using Maximum Likelihood (ML) and Bayesian Inference (BI) methods.The topology of the phylogenetic trees obtained by these two methods was similar',But in the result part,there was no detailed explanation of the difference between the two methods,and the Figure 7 does not indicate which tree was obtained by which method,Please explain the above issues.
3.'ModelFinder was employed to determine the best- fitting model for ML analysis.
ModelFinder was employed to determine the best- fitting model for ML analysis.'This need detailed explanation Comments on the Quality of English LanguageMinor editing of English language required
Author Response
Dear Reviewers:
Thank you for your letter and for the reviewers’ comments concerning our manuscript entitled “Analysis of the Chloroplast Genomes and Angiosperms353 Dataset of Three Carthamus Species Provides Insights into the Evolution of the Genus Carthamus” (ijms-2639957). Those comments are all valuable and very helpful for revising and improving our paper, as well as the important guiding significance to our researches. We have studied comments carefully and have made correction which we hope meet with approval. Questions are indicated by Q and responses are indicated by A. Modified sections in the article are shown in yellow.
Based on the reviewers' suggestions and the core content of this paper, we decided to change the article title. Changed to “Chloroplast Genomes and Phylogenetic Analysis of Three Carthamus (Asteraceae) Species”. In addition, we have optimized and abridged the full text, and the newly revised version differs from the earlier version.
Reviewer #3:
Q1. Why not use the whole cp genomes for phylogenetic analysis? Suggest using whole genome data to analyze the evolutionary tree.
A1. Ok, we used whole genome sequences to construct a phylogenetic tree, but this result is consistent with using protein-coding genes.
Q2. 'The phylogenetic trees were constructed using Maximum Likelihood (ML) and Bayesian Inference (BI) methods. The topology of the phylogenetic trees obtained by these two methods was similar',But in the result part,there was no detailed explanation of the difference between the two methods,and the Figure 7 does not indicate which tree was obtained by which method,Please explain the above issues.
A2. Ok, we've made the changes. Because the topologies obtained by the two methods agree, the phylogenetic tree in Figure combines the results of the Bayesian inference and maximum likelihood methods. The values on the nodes indicate the ML bootstrap values (left) and BI posterior probabilities (right). “Phylogenetic trees of the tribe Cardueae using Maximum likelihood (ML) and Bayesian inference (BI) analyses. (a) Phylogenetic tree of cpPCGs matrix; (b) Phylogenetic tree of nrITS matrix; (c) Phylogenetic tree of AP353 matrix. The values on the nodes indicate the ML bootstrap values (left) and BI posterior probabilities (right).”
Q3. 'ModelFinder was employed to determine the best-fitting model for ML analysis.
ModelFinder was employed to determine the best-fitting model for ML analysis.' This needs detailed explanation
A3. Ok, we've made the changes. Choosing a suitable model is more conducive to improving phylogenetic support and resolution. In IQ-TREE, there can be 2 ways for model evaluation, one is ModelFinder and the other is jModelTest. ModelFinder has more models than jModelTest, which is more favorable for model evaluation. The parameters for this run are iqtree2 -s seq.fas -m MFP -bb 1000 -nt 30.
Q4. Minor editing of English language required.
A4. Ok, we've made the changes.
Thanks for your reconsideration.
Yours sincerely,
Tiange Yang (E-mail: [email protected])
19 Oct., 2023
South-Central Minzu University
Reviewer 2 Report
Comments and Suggestions for Authors
IJMS Review
In this paper the authors report complete chloroplast genome sequences from 3 species of the genus Carthamus and comment on a number of genome structure features. The main analysis is a phylogenetic reconstruction which indicates that the 3 Carthamus species are monophyletic.
Overall, there is very little reported in this paper despite the long length of the manuscript. There are lengthy discussions of features such as GC content, repeat sequences, IR boundaries and codon usage, but none of it provides anything new to the literature and is of extremely little value. Nor do the details discussed play any role in the phylogenetic analysis. Since they cover features of the chloroplast genome that have been covered more than extensively in the literature for decades it does not seem necessary to include any of these things in this paper.
More importantly, the codon usage analysis has a serious problem. CAI values are reported on line 214 of page 8. However, CAI is a measure of usage relative to an assumed adaptive pattern – it needs a reference set of codon fitness values. When Sharp and Li introduced this in bacteria they used the codon usage of a few very highly expressed genes to generate fitness values. This allowed them to measure the degree of adaptiveness of other genes – essentially a measure of use of the presumed high fitness codons. Any time CAI is used you need a reference set of genes for fitness estimates and there needs to be a biological basis for using this set. You cannot simply choose a random set of genes and then measure CAI. Yet the reference set is never mentioned in the Materials and Methods of this paper so it is unclear what these CAI values mean. In the plant chloroplast genome the one gene that has been used as a reference is psbA and the authors should look at this literature and think about how they are using CAI. Right now it appears that they do not understand what they are doing and the CAI values they report are probably completely meaningless. To be honest I think that the whole analysis should simply be deleted since it is of no value whatsoever.
The analysis of adaptative evolution is also a bit confusing. It seems like sequences were fed into a program and the results then reported without actually looking at them or thinking about them in any meaningful way. Based on Figure 3 it seems like most regions of the genomes have about 3 variable sites in any window of 600 bases. This is extremely small, as expected given the conserved nature of the chloroplast genome – there is rarely much variation within a genus. It is also too small to do any meaningful Ka/Ks analysis. Before the results are taken seriously we need to see the actual Ka and Ks values – along with errors – so that they can be assessed. In addition the alignments should be considered carefully. Given the level of sequence conservation, the results for the accD gene could be the result of just one or two amino acid changes. I would need to see the alignment and what amino acid changes have occurred in accD before I assess the role of selection. Also, in section 2.6 were the results corrected for multiple tests? This is related to the previous point since it is likely that there are large errors around the Ka and Ks values so simply performing hundreds of tests and pulling out a positive result is not the correct way to go about things. Overall this analysis presented is simply too glib.
On lines 290-294 the authors make a curious comment that the IR is ‘self-replicating’. Of course, that is not true. Perhaps they are referring to the crossing over ‘correction’ that can occur between the two copies, which lowers the substitution rate. This needs to be clarified. As for the suggestion that a shift in the boundaries is an adaptive mechanism, there is absolutely no evidence for this statement and it makes no evolutionary sense. Really, the whole discussion of IR boundaries is pointless and should be removed.
I recommend that the paper be shortened significantly. The work should focus on the phylogenetic analyses and the other problematic, and pointless, analyses of genome sequence should be deleted entirely.
Author Response
Dear Reviewers:
Thank you for your letter and for the reviewers’ comments concerning our manuscript entitled “Analysis of the Chloroplast Genomes and Angiosperms353 Dataset of Three Carthamus Species Provides Insights into the Evolution of the Genus Carthamus” (ijms-2639957). Those comments are all valuable and very helpful for revising and improving our paper, as well as the important guiding significance to our researches. We have studied comments carefully and have made correction which we hope meet with approval. Questions are indicated by Q and responses are indicated by A. Modified sections in the article are shown in yellow.
Based on the reviewers' suggestions and the core content of this paper, we decided to change the article title. Changed to “Chloroplast Genomes and Phylogenetic Analysis of Three Carthamus (Asteraceae) Species”. In addition, we have optimized and abridged the full text, and the newly revised version differs from the earlier version.
Reviewer #1:
Q1. In this paper the authors report complete chloroplast genome sequences from 3 species of the genus Carthamus and comment on a number of genome structure features. The main analysis is a phylogenetic reconstruction which indicates that the 3 Carthamus species are monophyletic.
Overall, there is very little reported in this paper despite the long length of the manuscript. There are lengthy discussions of features such as GC content, repeat sequences, IR boundaries and codon usage, but none of it provides anything new to the literature and is of extremely little value. Nor do the details discussed play any role in the phylogenetic analysis. Since they cover features of the chloroplast genome that have been covered more than extensively in the literature for decades it does not seem necessary to include any of these things in this paper.
More importantly, the codon usage analysis has a serious problem. CAI values are reported on line 214 of page 8. However, CAI is a measure of usage relative to an assumed adaptive pattern – it needs a reference set of codon fitness values. When Sharp and Li introduced this in bacteria, they used the codon usage of a few very highly expressed genes to generate fitness values. This allowed them to measure the degree of adaptiveness of other genes – essentially a measure of use of the presumed high fitness codons. Any time CAI is used you need a reference set of genes for fitness estimates and there needs to be a biological basis for using this set. You cannot simply choose a random set of genes and then measure CAI. Yet the reference set is never mentioned in the Materials and Methods of this paper so it is unclear what these CAI values mean. In the plant chloroplast genome, the one gene that has been used as a reference is psbA and the authors should look at this literature and think about how they are using CAI. Right now it appears that they do not understand what they are doing and the CAI values they report are probably completely meaningless. To be honest I think that the whole analysis should simply be deleted since it is of no value whatsoever.
The analysis of adaptative evolution is also a bit confusing. It seems like sequences were fed into a program and the results then reported without actually looking at them or thinking about them in any meaningful way. Based on Figure 3 it seems like most regions of the genomes have about 3 variable sites in any window of 600 bases. This is extremely small, as expected given the conserved nature of the chloroplast genome – there is rarely much variation within a genus. It is also too small to do any meaningful Ka/Ks analysis. Before the results are taken seriously, we need to see the actual Ka and Ks values – along with errors – so that they can be assessed. In addition, the alignments should be considered carefully. Given the level of sequence conservation, the results for the accD gene could be the result of just one or two amino acid changes. I would need to see the alignment and what amino acid changes have occurred in accD before I assess the role of selection. Also, in section 2.6 were the results corrected for multiple tests? This is related to the previous point since it is likely that there are large errors around the Ka and Ks values so simply performing hundreds of tests and pulling out a positive result is not the correct way to go about things. Overall, this analysis presented is simply too glib.
On lines 290-294 the authors make a curious comment that the IR is ‘self-replicating’. Of course, that is not true. Perhaps they are referring to the crossing over ‘correction’ that can occur between the two copies, which lowers the substitution rate. This needs to be clarified. As for the suggestion that a shift in the boundaries is an adaptive mechanism, there is absolutely no evidence for this statement and it makes no evolutionary sense. Really, the whole discussion of IR boundaries is pointless and should be removed.
I recommend that the paper be shortened significantly. The work should focus on the phylogenetic analyses and the other problematic, and pointless, analyses of genome sequence should be deleted entirely.
A1. We thank the reviewers for their valuable suggestions and we have revised them accordingly. Based on these comments, we have shortened the length of the article and removed sections of little significance from the article, such as nucleotide diversity analysis, Ka/Ks analysis, CAI and ENC analysis of codons. We focus on presenting the chloroplast characteristics of the three Saffron species and their phylogenetic positions.
Thanks for your reconsideration.
Yours sincerely,
Tiange Yang (E-mail: [email protected])
19 Oct., 2023
South-Central Minzu University
Reviewer 3 Report
Comments and Suggestions for Authors
My comments are included in the attached file.

Author Response
Dear Reviewers:
Thank you for your letter and for the reviewers’ comments concerning our manuscript entitled “Analysis of the Chloroplast Genomes and Angiosperms353 Dataset of Three Carthamus Species Provides Insights into the Evolution of the Genus Carthamus” (ijms-2639957). Those comments are all valuable and very helpful for revising and improving our paper, as well as the important guiding significance to our researches. We have studied comments carefully and have made correction which we hope meet with approval. Questions are indicated by Q and responses are indicated by A. Modified sections in the article are shown in yellow.
Based on the reviewers' suggestions and the core content of this paper, we decided to change the article title. Changed to “Chloroplast Genomes and Phylogenetic Analysis of Three Carthamus (Asteraceae) Species”. In addition, we have optimized and abridged the full text, and the newly revised version differs from the earlier version.
Reviewer #2:
Q1. The reviewed paper reports the complete sequences of the chloroplast genome of three Carthamus species: C. lanatus, C. persicus and C. tinctorius. The complete chloroplast genome sequences were characterized and used for comparative and evolutionary studies. The applied methodology, high-throughput sequencing, provides high-quality data with a number of applications. Here, the Authors not only report the complete chloroplast genome of C. lanatus, C. persicus and C. tinctorius, but also based on the molecular data verified the systematic relationships between the studied species and other representatives of the tribe Cardueae. Furthermore, they identified plastome regions characterized with the highest variation and proposed them as potential DNA markers for further studies on the genus Carthamus. The observations included in the reviewed manuscript may become a valuable element of the discussion not only in the case of studies on the evolution and diversity of the genus Carthamus but also for other closely related taxa.
The objective of the paper is clear and unambiguous. The article is well structured, the methodology is correct and suitable for the realization of the paper’s objectives. However, a few improvements are needed. More exhaustive/detailed comments are incorporated into the manuscript file as notes. As I am not an English native speaker, I tried to focus only on elements that were important to the merit of the paper, not the English language or style. Please treat my comments/suggestions concerning the language as non-exhaustive.
A1. Ok, we've revised and optimized the full article accordingly.
Q2. In scientific works, the authority for a Latin species name is usually given, at least when it is first mentioned, e.g., Carthamus lanatus L. ("L." the abbreviation used for "Linnaeus"). Please apply it also for the other species mentioned in your manuscript.
A2. Ok, we've made the changes.
Q3. L. 18 Please explain why you resequenced the cp genome of C. tinctorius. So far, in GenBank, several accessions contain this species' complete plastome sequence, i.e., NC_030783, MZ779038, MZ779039, MK983238, KX822074. Did you perform any comparative study between your sequence and those published previously? Why didn't you take these sequences into account in your analyses?
A3. Thank you very much for your suggestion, we've made the changes. The resequencing of C. tinctorius was carried out mainly due to differences in cultivation areas and varieties from those already published. However, following suggestions from several reviewers, we have combined these published C. tinctorius chloroplast genomes for comparative intraspecific analyses.
Q4. L. 20-21, I suggest replacing “IR boundary changes” with “IR boundary locations”
A4. Ok, we've made the changes.
Q5. L.37, Harness? I think that “bind” will be better here.
A5. Ok, we've made the changes.
Q6. L. 40, Please improve that statement as chloroplasts are uniparentally inherited but depending on the plant group they could be maternally- or paternally inherited.
Chloroplast are maternally inherited in most angiosperms but in gymnosperms, the paternal inheritance is observed. Moreover, there are many examples for which, at least occasionally, biparental inheritance is reported.
A6. Ok, we've made the changes. The chloroplast genome has served as a valuable source of molecular data for investigating plant phylogeny and evolution. This is attributed to its predominantly ma-ternal inheritance, although patrilineal or biparental inheritance can occur in certain species, and its relatively conserved nature, which differs from plant mitochondrial genomes.
Q7. L. 42, Again, please improve that statement as the quadripartite structure is not the only possible scheme of the structure of the plastid genome
A7. Ok, we've made the changes. In most plants, the cp genomes possess a cyclic quadripartite structure with two single-copy regions separated by two inverted repeat regions, although linear structures are found in some species.
Q8. L. 50. This statement “...organelle genomes are inherited maternally...” is only partially true. See my comment for L.40
A8. Ok, we've made the changes. Unlike large nuclear genomes, angiosperm organelle genomes are predominantly maternally inherited and are prone to high rates of genetic drift in small populations, resulting in potentially less accurate estimates of lineage divergence
Q9. L. 52-57
These sentences are almost identical to the original text:
McDonnell AJ, Baker WJ, Dodsworth S, Forest F, Graham SW, Johnson MG, Pokorny L, Tate J, Wicke S, Wickett NJ. Exploring Angiosperms353: Developing and applying a universal toolkit for flowering plant phylogenomics. Appl Plant Sci. 2021 Jul 26;9(7):10.1002/aps3.11443. doi: 10.1002/aps3.
Please try to write it in your own words.
A9. Ok, we've made the changes. The Angiosperms353 dataset has gained widespread usage in angiosperm phylogeny [6]. This dataset comprises 353 targeted low-copy nuclear genes that are highly repre-sentative and have become a valuable tool for molecular systematics and population genetics [6-8]. Additionally, researchers can easily access the dataset through enrich-ment sequencing (Hyb-Seq), transcriptome sequencing (RNA-Seq), and genome skimming to obtain the corresponding data [9]. Previous studies have demonstrated the feasibility of applying the angiosperms353 dataset in phylogenetic studies of Asteraceae [10].
Q10. L. 70, Remove a dot after the (2n=24)
A10. Ok, we've made the changes.
Q11. L. 81-83, reference to the literature should be improved as it is confusing here whether Hanelt (1963) or Estilai and Knowles (1976) is reference no. [19]
A11. Ok, we've made the changes.
Q12. L. 95-97, The sentence needs improvement as it sounds strange that the Carduncellus-Carthamus complex can be divided into one genus. The expression “divide” suggests that something will be split so at least two elements should be observed after dividing/splitting.
A12. Ok, we've made the changes.
Q13. L. 98, nrITS is used here for the first time so the abbreviation should be explained
A13. Ok, we've made the changes. nrITS (nuclear ribosomal internal transcribed spacer)
Q14. L. 108, Combined? I do not follow your concept here. Just write that you performed a comparative analysis.
A14. Ok, we've made the changes.
Q15. L. 129, Replace “six ATP synthases” with “six subunits of ATP synthase”
A15. Ok, we've made the changes.
Q16. L. 144-155, When talking about SSRs please use the expression mono-, di, tri, tetra penta or hexanucleotide REPEATS when talking about the motifs that are observed for these genomic elements.
A16. Ok, we've made the changes.
Q17. L. 156
Did you mean here “intergenic spacer”? Improve it.
A17. Ok, we've made the changes.
Q18. L. 170
You have used mVISTA tool here, that is true, but name exactly what kind of analysis you performed here as the expression “mVISTA analysis” is not the right choice.
A18. Ok, we've made the changes. These SSRs were mainly distributed in the intergenic spacer regions.
Q19. L. 174-176
Please be more specific when writing about the level of divergence for these regions – at least give the threshold value for Pi value you have used to distinguish these regions.
A19. Ok, we've made the changes. Since there is so little data on chloroplasts in the genus Carthamus, we removed the section on nucleotide diversity analysis based on the advice of two other reviewers, as it was not of much significance or value.
Q20. L. 185, I think that you should write here that you tried to identify possible contraction and expansion of the IR regions or simply that you identified the location of the SSC/IR and LSC/IR boundaries.
A20. Ok, we've made the changes.
Q21. L. 185-193, The description of the location of the SSC/IR and LSC/IR boundaries should be improved:
* I think that the LSC-IRb boundary is located within sequences for rps19 genes not rps19 genes were located in the LSC-IRb boundary
* IRa-SSC boundary is a line/junction between two cp genome regions thus it cannot contain any sequences (e.g. trnN and ndhF genes or rpl2 and trnH genes)
* in one place you write “IRa-SSC boundary contained trnN and ndhF genes” and a few lines later “SSC-IRa boundary had rpl2 and trnH genes” – please check it
A21. Ok, we've made the changes. The trnN and rpl2 genes were located within the IR regions, ndhF genes were located with-in the SSC regions, and the LSC-IRb and SSC-IRa boundaries were located on the ycf1 and rps19 genes, respectively.
Q22. L.206
See my comment for lines 389-390
A22. Ok, we've made the changes.
Q23. L. 216
Replace “a rare” with “the less frequent” or “the rarest”
A23. Ok, we've made the changes.
Q24. L. 216-218
“encoded the most frequent occurrence of” the least frequent occurrence of”?
Just simply write that XXX codon encoding YYY was the most common or the last frequent, respectively.
A24. Ok, we've made the changes. The leucine was found to be the most abundant amino acid in these cp genomes, while cysteine was the less frequent amino acid.
Q25. L. 220
Tryptophan and methionine are coded by one codon so there is no sense in using the expression “by only one codon”
A25. Ok, we've made the changes. Both Tryptophan (UGG) and methionine (AUG) were encoded by a single codon and had RSCU values of 1.
Q26. L. 233-234
Sentence needs improvement. You should write that you performed the analysis for selected representatives of the genus Carthamus and Centaurea and that you describe results obtained for 15 species’ combinations.
A26. Ok, we've made the changes. Since there is so little data on chloroplasts in the genus Carthamus, we removed the section on Ka/Ks analysis based on the advice of two other reviewers, as it was not of much significance or value.
Q27. L.243
Replace “conservation of these genes” with “conservative character of these genes”
A27. Ok, we've made the changes.
Q28. L. 246
Certain? Please be more specific.
A28. Ok, we've made the changes.
Q29. L. 253
Conducted?
A29. Ok, we've made the changes. We performed informative loci and tree-building modeling statistics for the three sequence matrices (Table S8).
Q30. L. 261-262
The phylogenetic tree based on nrITS matric shows also close relationship between Carthamus and Centaurea
A30. Ok, we've made the changes. Nevertheless, the overall structure remained similar across most branches of the three phylogenetic trees. These trees indicated that the Carthamus-Carduncellus complex was monophyletically sister to Centaurea in the Centaureinae branch.
Q31. L. 291
Please give reference for the statement “IR region is known to be self-replicating”
A31. Ok, we've made the changes. Based on the suggestions of two other reviewers, we removed this sentence
Q32. L. 292-293
This statement “generally lower Ks value for IR genes than for SC genes” is based on the
cited reference. Why did you mention it here? Did you also observe the same phenomena in
your data?
A32. Ok, we've made the changes. We removed this sentence.
Q33. L. 302-303
Actually, you should use the opposite order: further studies are needed to verify your
hypothesis.
A33. Ok, we've made the changes.
Q34. L. 310
I think it will be beneficial to Replace “process of adapting to adversity” with “process of
adaptation to unfavorable conditions” or “process of adaptation to abiotic stress”
A34. Ok, we've made the changes.
Q35. L. 319
Use the italic font for the gene’s name/symbols
A35. Ok, we've made the changes.
Q36. L. 326-327
Sentence needs improvement
Below I give my suggestion which can be used to rephrase your statement.
These positively selected genes may be linked/associated(?) with the unique edaphic and microclimatic conditions (are there any?) of habitats of Carthamus and Cardueae species.
A36. Ok, we've made the changes.
Q37. L. 338-341
Yes, all these factors can be responsible for observed differences but do not forget about the abundance of the AP353 data in comparison to cpPCGs. Just a brief look at Table 2 explains everything. Analysis of a high number of sites that evolve according to different scenarios may lead to different directions/results.
Convergent evolution? In this sentence, you are explaining the reasons for inconsistency in results derived from two sets of data (nuclear and chloroplast) so I believe the “convergent” does not fit here.
A37. Ok, we've made the changes. Differences between the plastid phylogenetic tree and the nuclear phylogenetic tree may arise from various factors, including hybridization, incomplete lineage sorting (ILS), chloroplast capture, and genetic differences in plastids
Q38. L. 341-345
These two sentences actually repeat the same information.
A38. Ok, we've made the changes.
Q39. L. 355-356
I think that this hypothesis will be hard to defend. Based on the presented results only Figure 7a may suggest that (the position of Centaurea cyanus in comparison to other representatives of the genus Centaurea and Carthamus), but this is very weak evidence as no other sets of data confirmed that.
If you want to discuss the divergence between the genus Centaurea and Carthamus further studies will be needed in which a higher number of taxa representing these two genera should be used. Nevertheless, as you are trying to discuss this issue, I highly recommend enhancing your studies with divergence time estimation (see also my comment for L. 412)
A39. Ok, we've made the changes. The timing of the differentiation of the Asteraceae has been studied previously, and it has also been shown that Centaurea differentiates earlier than Carthamus. However, based on the suggestions of two reviewers, we decided to remove this part of the discussion and favor phylogeny more.
Q40. L. 365
Replace “These leaves were collected” with “Fresh leaves of these species were collected”
A40. Ok, we've made the changes.
Q41. L. 378-380
GetOrganelle was developed for assembling organellar genomes so did you know any other case of studies in which the GetOrganelle program was used for assembling non-organellar regions? Did you validate the obtained ITS sequences with another approach? What sequence did you use as a reference during the ITS sequence annotation in Geneious software?
A41. Ok, we've made the changes. Assembly of nrDNA using GetOrganelle has been common, as reported in other articles, and the sequences assembled are largely consistent with the results of one generation of sequencing. Such as “GetOrganelle: a fast and versatile toolkit for accurate de novo assembly of organelle genomes” https://github.com/Kinggerm/GetOrganelle; “New insights into infrageneric relationships of Lonicera (Caprifoliaceae) as revealed by nuclear ribosomal DNA cistron data and plastid phylogenomics. https://doi.org/10.1111/jse.13014”. For the annotation, we based on the published nrDNA sequences of Carthamus tinctorius (KY397481) in NCBI as reference to annotate rrn18S, rrn5.8S, rrn26S.
Q42. L. 382
Replace “misa” with “MISA”
A42. Ok, we've made the changes.
Q43. L. 384-387
Improve that sentence. This sentence should inform that you have used REPuter software to
identify four types of repeats (forward, reverse, complement and palindromic).
A43. Ok, we've made the changes. We used REPuter (https://bibiserv.cebitec.uni-bielefeld.de/reputer) to predict the palindromic and complement, forward and reverse repeats.
Q44. L. 387
The sentence should be rephrased. Below I give my suggestion:
The minimum repeat size and Hamming distance were set to 30 and 3, respectively.
A44. Ok, we've made the changes. The minimum repeat size and Hamming distance were set to 30 and 3, respectively.
Q45. L.389-390
Can you give any reference here that will support your decision, i.e. to apply the approach that removes PCGs shorter than 300 bp from this analysis?
A45. Ok, we've made the changes. This method has been reported before, for example,
Wright, F., The ‘effective number of codons’ used in a gene. Gene 1990, 87, (1), 23-29. DOI: 10.1016/0378-1119(90)90491-9;
Zhou, M.; Li, X., Analysis of synonymous codon usage patterns in different plant mitochondrial genomes. Molecular Biology Reports 2009, 36, (8), 2039-2046. DOI: 10.1007/s11033-008-9414-1
Q46. L.391
* Reference is missing for CodonW v.1.4.4
* Replace “1rd” with “1st"
A46. Ok, we've made the changes.
Q47. L. 392
Replace “2rd” with “2nd"
A47. Ok, we've made the changes.
Q48. L. 398
Reference is missing for mVISTA software
A48. Ok, we've made the changes.
Q49. L. 400
Reference is missing for MAFFT software (I find it further below with ref. no. 64, but here MAFFT software is mentioned for the first time).
A49. Ok, we've made the changes.
Q50. L. 407-411
Reference is missing for Ka/Ks values interpretation
A50. Ok, we've made the changes.
Q51. L.412
As your paper refers to the evolution of genus Carthamus (Title) and in the discussion you devote an entire section to evolution within the Cardueae tribe I highly recommend adding the divergence time estimation to the phylogenetic analyses.
A51. Ok, we've made the changes. Based on the reviewers' suggestions and the core content of this paper, we decided to change the article title. Changed to “Chloroplast Genomes and Phylogenetic Analysis of Three Carthamus (Asteraceae) Species”. The timing of differentiation in Asteraceae has been studied previously (Phylotranscriptomic insights into Asteraceae diversity, polyploidy, and morphological innovation; https://doi.org/10.1111/jipb.13078) and we decided not to add this section.
Q52. L.414
Give the full species name for the C. indicum (Chrysanthemum indicum) as it is used here for the first time and to avoid confusion.
A52. Ok, we've made the changes.
Q53. L. 439
Replace “higher level” with “high level”
A53. Ok, we've made the changes.
Q54. L. 441
Replace “the boundaries between” with “the boundaries location between”
A54. Ok, we've made the changes.
Q55. L. 451
Seem my comment for L. 355-356
A55. Ok, we've made the changes.
Q56. Table 1
Replace “Genbank number” with “Genbank accession number”
A56. Ok, we've made the changes.
Q57. Figure 1
Low quality. The fonts are blurred and too small to read.
A57. Ok, we've made the changes.
Q58. Figure 2
In the case of Figure 2c the fonts are blurred and too small to read.
Moreover, there is no information about Figure 2c in the Figure 2 caption.
A58. Ok, we've made the changes.
Q59. Figure 3 caption
The last sentence here is rather an element of the Materials and Methods section. It can be
removed from here.
A59. Ok, we've made the changes.
Q60. Figure 4 caption
* “chaining relationships”?
* “The number of bp represented by the arrow showed genes away from a specific region of
the chloroplast genome”?
A60. Ok, we've made the changes.
Q61. Figure 5 caption
Codon content? Perhaps rather “codon frequency”?
A61. Ok, we've made the changes.
Q62. Figure S1 caption
* replace “MAUVE alignment of Carthamus” with “MAUVE alignment of chloroplast
genomes of Carthamus ”
* long squares? Squares could be long?
* short squares? Squares could be short?
A62. Ok, we've made the changes.
Q63. Figure S2 caption
The caption should be improved as it shows the heat-map comparing the Ka/Ks ratios for
XX(72?) cpPCGs shared by 19 Cardueae species. The blue shades denote Ka/Ks<1, the red
shades represent Ka/Ks>1, gray color indicates ...???
A63. Ok, we've made the changes.
Q64. Table S1 caption
“Phylogenetic analysis of all species and their NCBI accession numbers in the article.”
The caption needs improvement as it contains 20 species which are not all species in the
article (AP353 matrix?)!
A64. Ok, we've made the changes.
Q65. Table S5 caption
“Position information annotation” ??? Not clear
A65. Ok, we've made the changes. Table S5: Distribution region of long repeats for four Carthamus species.
Q66. Table S8 caption
Caption needs improvement:
Ka/Ks ratios for 79 cpPCGs shared by the selected Carthamus and Centaurea representatives
A66. Ok, we've made the changes.
Q67. Table S9 caption
Caption needs improvement:
Ka/Ks ratios for 72 cpPCGs shared by 19 species of the tribe Cardueae.
A67. Ok, we've made the changes.
Thanks for your reconsideration.
Yours sincerely,
Tiange Yang (E-mail: [email protected])
19 Oct., 2023
South-Central Minzu University